# Factors associated with vaccine coverage improvements in Senegal between 2005 and 2019: a quantitative retrospective analysis

Hannah K Smalley,[1] Francisco Castillo-Zunino,[1] Pinar Keskinocak ,[1,2] Dima Nazzal,[1] Zoë M Sakas,[3] Moussa Sarr ,[4] Matthew C Freeman [3]

[1]H. Milton Stewart School of Industrial and Systems Engineering, Georgia Institute of Technology, Atlanta, Georgia, USA
[2]Center for Health and Humanitarian Systems, Georgia Institute of Technology, Atlanta, Georgia, USA
[3]Gangarosa Department of Environmental Health, Rollins School of Public Health, Emory University, Atlanta, Georgia, USA
[4]Institut de Recherche en Santé de Surveillance Epidemiologique et de Formation, Dakar, Senegal

**Correspondence to**
Dr Hannah K Smalley;
hannahsmalley@gatech.edu

## ABSTRACT

**Objective** Senegal has demonstrated catalytic improvements in national coverage rates for early childhood vaccination, despite lower development assistance for childhood vaccines in Senegal compared with other low-income and lower-middle income countries. Understanding factors associated with historical changes in childhood vaccine coverage in Senegal, as well as heterogeneities across its 14 regions, can highlight effective practices that might be adapted to improve vaccine coverage elsewhere.

**Design** Childhood vaccination coverage rates, demographic information and health system characteristics were identified from Senegal's Demographic and Health Surveys (DHS) and Senegal national reports for years 2005–2019. Multivariate logistic and linear regression analyses were performed to determine statistical associations of demographic and health system characteristics with respect to childhood vaccination coverage rates.

**Setting** The 14 administrative regions of Senegal were chosen for analysis.

**Participants** DHS women's survey respondents with living children aged 12–23 months for survey years 2005–2019.

**Outcome measures** Immunisation with the third dose of the diphtheria-tetanus-pertussis vaccine (DTP3), widely used as a proxy for estimating immunisation coverage levels and the retention of children in the vaccine programme.

**Results** Factors associated with childhood vaccination coverage include urban residence (β=0.61, p=0.0157), female literacy (β=1.11, p=0.0007), skilled prenatal care (β=1.80, p<0.0001) and self-reported ease of access to care when sick, considering travel distance to a healthcare facility (β=−0.70, p=0.0009) and concerns over travelling alone (β=−1.08, p<0.0001). Higher coverage with less variability over time was reported in urban areas near the capital and the coast (p=0.076), with increased coverage in recent years in more rural and landlocked areas.

**Conclusions** Childhood vaccination was more likely among children whose mothers had higher literacy, received skilled prenatal care and had perceived ease of access to care when sick. Overall, vaccination coverage is high in Senegal and disparities in coverage between regions have decreased significantly in recent years.

## STRENGTHS AND LIMITATIONS OF THIS STUDY

⇒ This study explains associations between diphtheria-tetanus-pertussis vaccine coverage and demographic and health system characteristics at the subnational level within Senegal.
⇒ This study evaluates changes in vaccination coverage over time by administrative region and geographical area.
⇒ Analysis was not performed at the district level and thus heterogeneities at the subregional level were not evaluated.

## BACKGROUND

Vaccination is one of the most cost-effective and influential public health interventions.[1] For each dollar spent on routine and supplementary vaccinations in low-income and middle-income countries from 2011 to 2020, the average country-level return on investment is estimated to be 44 times the cost.[2] High vaccination coverage is necessary for controlling, eliminating or eradicating vaccine-preventable diseases.[3] Gavi, The Vaccine Alliance (Gavi), was launched in 2000 and is one of the main sources of vaccine financing for eligible low-income and middle-income countries. Some but not all Gavi-supported countries were able to reach the 2015 Global Vaccine Action Plan goal of 90% coverage for the third dose of the diphtheria-tetanus-pertussis vaccine (DTP3) by 2010.[4 5]

In 2017, Africa reported a DTP3 coverage of 72%, the lowest percentage of all the WHO regions.[4] Within the African Region, several countries have outperformed their peers with regard to high and sustained vaccination coverage. DTP3 coverage in Senegal increased from 52% to 93%,[6] outperforming other low-income (LIC) and lower-middle income (LMIC) countries, despite lower

development assistance for health (DAH) relative to many other countries in this group.[7 8] The government of Senegal has demonstrated a commitment to improving access to healthcare, including childhood vaccination, which likely contributed to the country's success.[9–11] However, like other countries in sub-Saharan Africa, Senegal's childhood vaccine coverage was not consistent between Senegal's 14 administrative regions.

The purpose of this study was to identify factors associated with DTP3 coverage in Senegal at both the household level and regionally. DTP3 is often used as a proxy for estimating the retention of children in the vaccine programme.[3] Subnational trends in early childhood vaccine coverage often illustrate inequities in vaccine service delivery,[12 13] and looking at national-level data alone may overestimate vaccine system performance.[14] Factors statistically associated with full childhood immunisation coverage have been identified in previous studies.[15–18] We complement prior work by focusing on DTP3 coverage in Senegal, with an analysis of changes in coverage over time by administrative region from 2005 to 2019. We analysed geographical and population factors, including female literacy and access to skilled prenatal care. We also considered factors aggregated at the regional level including poverty and the number of healthcare workers.

## METHODS

We conducted a study to assess the demographic associations of early childhood vaccination within Senegal, including assessing the impact of heterogeneities at the subnational level (ie, between administration regions). For the study period of 2005–2019, we analysed geographical and population factors for each of Senegal's 14 administrative regions to examine potential associations with DTP3 coverage. The analysis was conducted as part of the Vaccine Exemplars Programme, the purpose of which was to identify how and why some countries have achieved better-than-average and sustained coverage of early childhood vaccines.[11 19–21]

### Data sources

Household-level immunisation data was collected from Senegal's Demographic and Health Surveys (DHS) (years 2005, 2010/2011, 2012/2013 and yearly from 2014 to 2019),[22] with data provided for all 14 regions and all years (notably 3 of the regions were created/split from existing regions in 2008; coverage for these regions in 2005 is estimated to be equal to that of the original combined regions). The data can be accessed on approval from the DHS Programme.[23] DHS surveys were sampled to be representative at the regional level, and thus we were not able to analyse statistical associations at the smaller district level. Twenty-eight sampling strata are created which correspond to urban and rural parts for each of the 14 regions; samples are drawn independently from these sampling strata. Specific details on sampling methods for

each survey year can be found in the appendices of each report.[22] Data extracted from the surveys include DTP3 vaccination (recommended administration to children at 14 weeks in Senegal), residence type (urban or rural), female literacy, proportion receiving prenatal care from skilled providers and self-reported problems accessing care when sick (ie, women were asked if distance, permission, money and travelling alone were significant barriers when seeking treatment).

Child immunisation is covered in the women's questionnaire in DHS surveys with responses representing each child of the mother surveyed. Literacy is assessed in the survey by the questioner asking the respondent to read from a card.[24] Prenatal (or antenatal) care is assessed by asking women who attended their last birth, with the following options: doctor, nurse/midwife, auxiliary nurse/midwife, community health worker, other health worker, traditional birth attendant, other and no antenatal care.

Region-specific population density and poverty (reported for year 2018) were determined from Senegal national reports.[25] Poverty incidence is estimated using the cost of basic needs approach, which considers the proportion of the population unable to meet their basic needs and combines a country-specific food poverty line and non-food poverty line. Angence Nationale de la Statistique et de la Démographie (ANSD) provides more details with respect to this calculation specific to Senegal.[26] We collected regional counts of skilled healthcare workers from the Ministère de la Santé et de l'Action sociale (2018).[27]

### Data analysis

Bivariate and multivariate logistic and linear regression analyses were performed to determine statistical associations of factors with respect to coverage of DTP3; factors included in multivariate analysis include residence type (urban or rural), female literacy, self-reported problems accessing care when sick and prenatal care provider. We calculated DTP3 coverage—the dependent variable in our models—for each region of Senegal using DHS data for each year available during the study period. The immunisation coverage for a region was calculated as the percentage of living children ages 12–23 months represented in the survey for that region who received DTP3. Data must be weighted because the overall probability of a household being selected is not constant across regions and residence types.[28] Thus, we compute the individual weight for women and households to adjust for differences in probability of selection and response between cases in a sample. We filter the data for living children ages 12–23 months. For the time series data, we do this for each year of data available. A vaccination is considered administered if it is reported by the mother or marked on the vaccination card with or without a date.

For our analysis, we consider a woman to be literate if she is able to read a whole sentence or part of a sentence as reported in the DHS survey.[24 28] We consider a woman

**Table 1** Senegal household survey data years and unweighted number of survey respondents meeting criteria for inclusion

| Survey year | Respondents fitting criteria |
| --- | --- |
| 2005 | 2138 |
| 2010/2011 | 2377 |
| 2012/2013 | 1329 |
| 2014 | 2662 |
| 2015 | 1310 |
| 2016 | 1323 |
| 2017 | 2390 |
| 2018 | 1337 |
| 2019 | 1183 |

to have received skilled prenatal care if she answered 'doctor', 'nurse/midwife' or 'auxiliary nurse/midwife' as attending her most recent birth.

We calculated the Pearson correlation coefficients (r) to determine statistically significant correlations between the quantitative factors and between factors and DTP3 coverage. We clustered regions geographically into two groups (coastal and inland), calculated the variance in DTP3 values over time for each region, and performed independent sample t-tests to compare these two groups of variances.

### Patient and public involvement

There was no patient or public involvement in the study design of this research, the interpretation of the results, or the writing or editing of this document. All data used in this analysis were deidentified public health data.

### RESULTS

Table 1 reports the number of DHS survey respondents (unweighted) in each survey data set which fit the criteria for inclusion (ie, with living children aged 12–23 months). The number of respondents in each region per year is reported in online supplemental table S1.

### Factors associated with DTP3 coverage by administrative region

For each region in Senegal, online supplemental table S2 reports the weighted percentage of survey respondents with 'yes' responses per household factor for year 2019; data are filtered for respondents who meet inclusion criteria. DTP3 coverage by region ranged from 73% to 100% in 2019 (figure 1, online supplemental table S2). Higher DTP3 coverage rates were observed in Dakar (the most populous region and includes the capital city) and its surrounding regions, and along the coast, as well as in the landlocked region of Matam. In 2018, there were approximately 6600 people per km$^2$ in Dakar, which is predominantly urban (97% urban).[29] The population density in other regions was at most 362 people per km$^2$.

Population density does not have a statistically significant correlation with DTP3 coverage at the regional level, though this correlation is higher when excluding Dakar (Pearson's correlation r=0.39, p=0.19).

Unadjusted bivariate analyses (figure 2) revealed significant associations (p=0.05) between higher DTP3 vaccination and the following:

► Positive associations:
  – Type of place of residence (urban) (β=0.61, p=0.0157).
  – Higher female literacy (β=1.11, p=0.0007).
  – Higher skilled prenatal care (β=1.80, p<0.0001).
► Negative associations:
  – More barriers when seeking treatment with respect to
    – Distance (β=−0.70, p=0.0009).
    – Travelling alone (β=−1.08, p<0.0001).

We find that significant associations hold for female literacy, skilled prenatal care, distance and travelling alone when adjusting by type of place of residence (urban).

Multivariate regression revealed correlations between factors (figure 2), with only three factors remaining statistically significant: positive associations with female literacy (β=0.92, p=0.0065) and skilled prenatal care (β=1.58, p<0.0001) and negative associations with more barriers when seeking treatment with respect to travelling alone (β=−0.82, p=0.0026) (more details provided in online supplemental table S3). We also found correlations between population density and fewer barriers when seeking treatment with respect to distance (r=0.61, p=0.02) and travelling alone (r=0.48, p=0.08); travelling alone is correlated with distance (r=0.51, p<0.0001).

Coastal regions, namely, Dakar and Ziguinchor, have the highest female literacy rates (68% and 70%, respectively). The percentage of women who received skilled prenatal care from a doctor, nurse or midwife was lowest in Tambacounda and Kedougou (88% and 85%, respectively), compared with over 95% in other regions (figure 1).

Poverty level did not statistically explain regional differences in DTP3 coverage (bivariate regression coefficient β=−0.17, p=0.157), potentially due to heterogeneity within regions. We also found no statistical associations between DTP3 coverage and the number of doctors, nurses and midwives per 100 000 population using bivariate or multivariate regression analysis (figure 3, more details provided in online supplemental table S4).

### Access to health services

Women in regions with low DTP3 coverage reported more barriers to accessing care with respect to (1) travelling alone to seek care and (2) distance from healthcare facilities. In the more remote, inland regions of Senegal, as many as 42% of female respondents in 2019 identified travelling alone to seek care as a significant barrier to accessing care. Distance to healthcare facilities was also a substantial barrier to accessing care in these regions of Senegal (identified as a barrier by 50% of

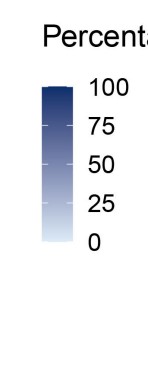

## (i) DTP3 coverage by region of Senegal (DHS 2019)

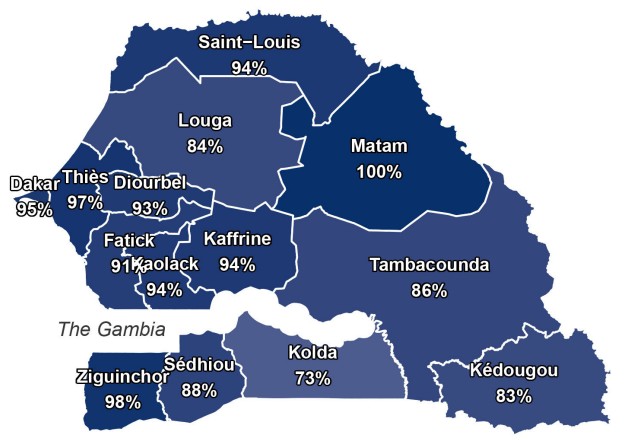

### (ii) Female population considered literate

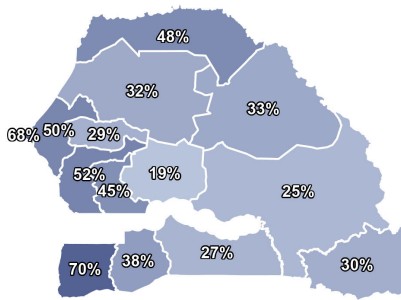

### (iii) Received skilled prenatal care

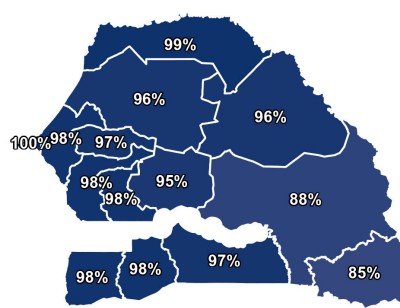

### (iv) Ease of access to care − Traveling alone

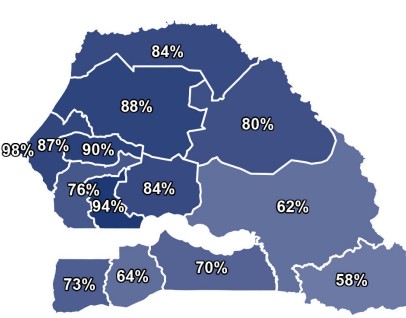

**Figure 1** DTP3 coverage by region of Senegal and regional proportions of select metrics.[30] 'Skilled prenatal care' refers to the proportion of women ages 15–49 who had a live birth within 5 years before the survey who had prenatal care provided by a skilled healthcare worker (2017). Skilled care includes doctor, nurse and midwife. Unskilled care includes matron/traditional midwife, 'other' or none. 'Ease of Access to Care-Travelling Alone' refers to the proportion of female survey respondents by region who did not consider travelling alone to be a significant problem with accessing care when sick (2017). DHS, Demographic and Health Survey; DTP3, third dose of diphtheria-tetanus-pertussis vaccine.

female respondents in Tambacounda, the largest region by geographical area). Sociodemographic characteristics of this survey group are reported in online supplemental table S5.[30]

### Coverage over time by administrative region and geographical location

DTP3 coverage from 2005 to 2019 by region in Senegal is reported in figure 4.[31] We observed that the DTP3 coverage was consistently higher for the west and southwestern regions located near Dakar and along the coast, and the variance of DTP3 coverage over time was consistently lower for this group of coastal regions (t-test, p=0.076) compared with the group of eastern and northern regions. The region with the lowest DTP3 coverage in 2019 was Kolda (73%), following a sharp drop in DTP3 coverage after 2017 compared with previous years. The rural and landlocked regions in the north and east demonstrated more variability in coverage over time. However, among these regions, Tambacounda and Kedougou saw large improvements, from 73% to 86%

and 60% to 83%, respectively between 2017 and 2019; notably, Matam saw the largest increase (from 73% to 100%) in coverage among all regions between 2010 and 2019.

### DISCUSSION

We conducted a study to assess the demographic associations of early childhood vaccination within Senegal. Observing heterogeneities in DTP3 coverage across Senegal and comparing those differences with demographic and geographical factors may aid decision-makers in addressing disparities in vaccination access and uptake. Our analysis of household level data revealed associations between DTP3 coverage and (1) female literacy, (2) access to skilled prenatal care and (3) challenges accessing care while travelling alone.

Early childhood vaccination in Senegal steadily increased for all recommended vaccines from 2002 to 2007, with coverage remaining mostly constant through

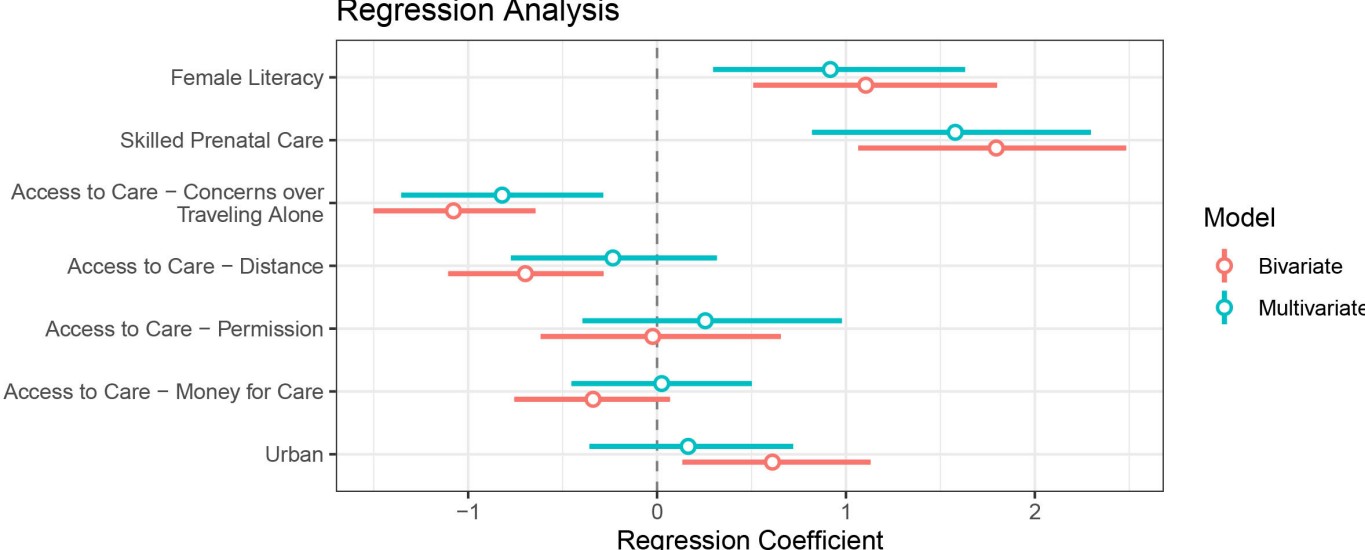

**Figure 2** Regression analysis for household-level factors associated with DTP3 vaccination by DHS 2019 survey response per living child aged 12–23 months (weighted N=1129). The bars correspond to 95% CI. Bars which do not intersect 0 correspond to statistical associations with DTP3 vaccination (<0 implies negative association, >0 implies positive association). DHS, Demographic and Health Survey; DTP3, third dose of diphtheria-tetanus-pertussis vaccine.

2019.[11] [32–37] Based on an analysis of qualitative data including interviews with Senegalese administrators at various levels and focus group discussions with caregivers and community health workers, Senegal's immunisation programme has been successful due to 'strong governance, collaboration, evidence-based decision-making, community ownership and an overall commitment to health and prioritisation of vaccine programming from all stakeholders and government officials'.[11] Understanding the existing disparities between regions in Senegal, alongside related factors, may support improvements in health equity through utilisation of Senegal's existing governance structures and strong community health worker programme.

We found that higher female literacy was significantly associated with higher DTP3 coverage. Female literacy and higher maternal education have been shown to be positively associated with higher vaccination coverage globally;[18] [38–40] it is atypical that Senegal, with low female literacy at the national level,[41] has relatively high DTP3

coverage compared with other LMICs. While mothers with high literacy and education may be more likely to independently seek vaccinations for their children,[42] the health system in Senegal has been able to reach the more vulnerable rural communities with lower female literacy through community health workers and community-based health huts; 'health huts are the backbone of Senegal's community health system, and they deserve much credit for Senegal's impressive progress on disease prevention and treatment, including its far-reaching immunisation programme'.[9] Involvement of community health workers such as respected female elders known as 'bajenu gox' have served as a link between the healthcare services (ie, health centres, health posts, health huts) and the communities, facilitating and coordinating immunisation outreach activities.[10]

We found that skilled prenatal care was associated with higher DTP3 coverage in Senegal, which aligns with existing literature. Access to skilled healthcare workers has an important role in childhood vaccination. Limited

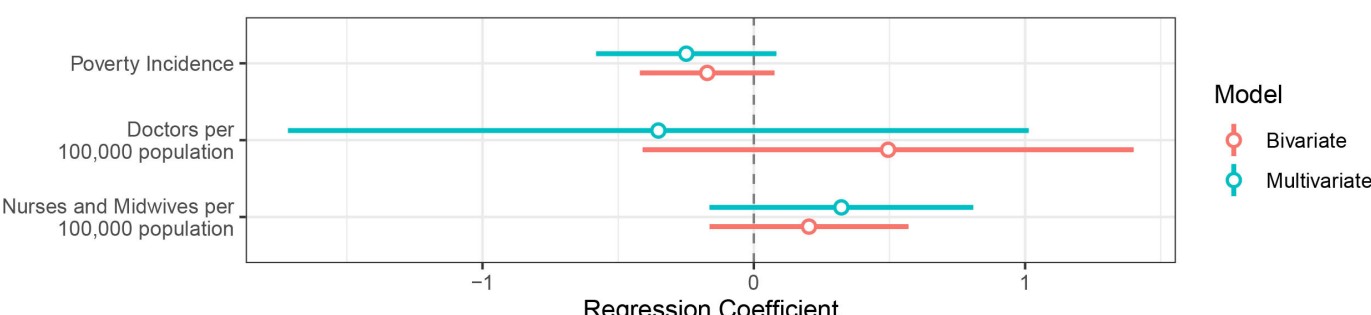

**Figure 3** Regression analysis for region-level factor associations with DTP3 vaccination. Poverty incidence is estimated using the cost of basic needs approach, which considers the proportion of the population unable to meet their basic needs and combines a country-specific food poverty line and non-food poverty line.[26] The bars correspond to 95% CIs. DTP3, third dose of diphtheria-tetanus-pertussis vaccine.

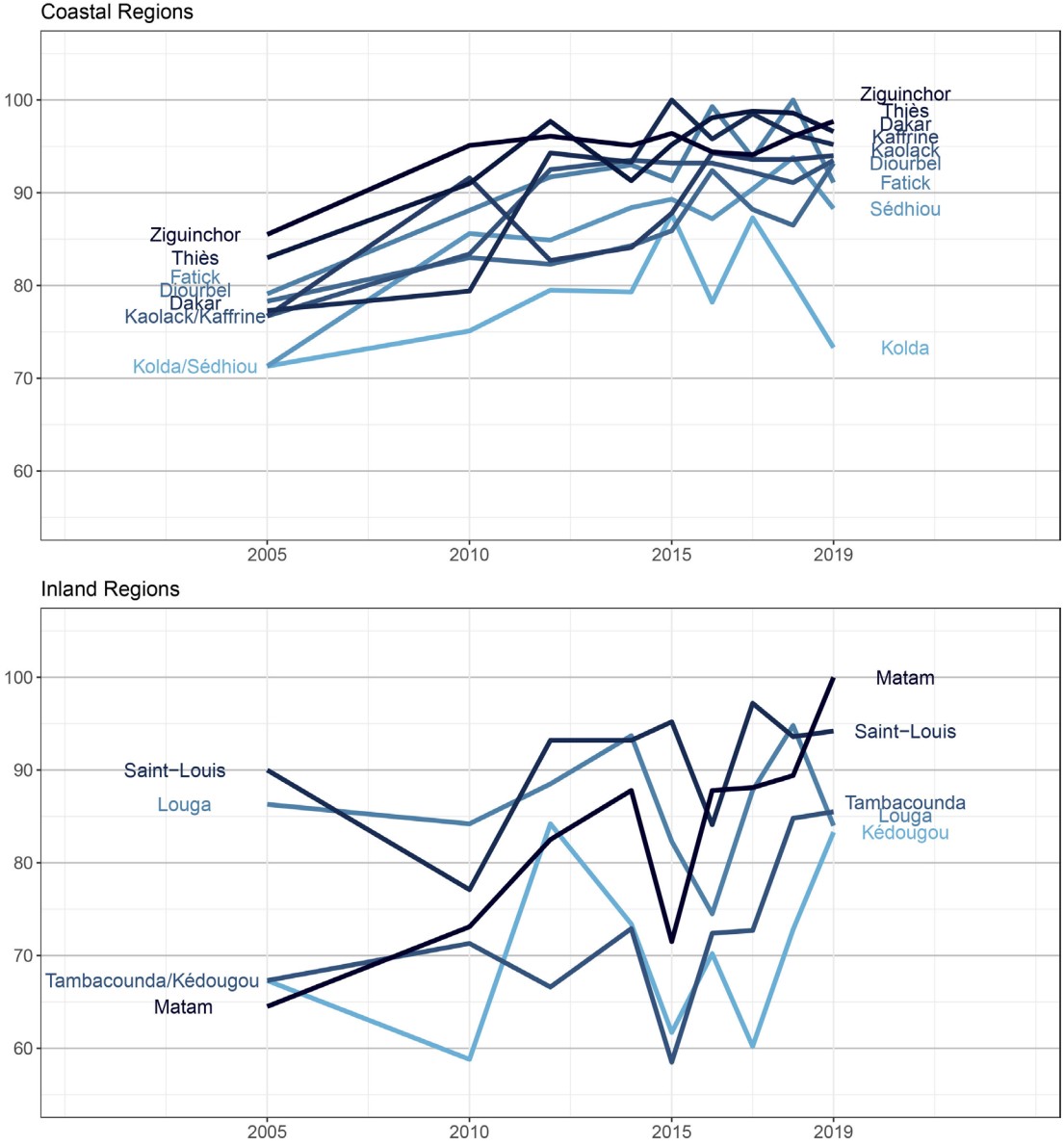

**Figure 4** DTP3 coverage in Senegal by year and region (DHS 2005–2019). The regions of Kaffrine, Sedhiou and Kedougou were split from existing regions in 2008.[31] DHS, Demographic and Health Survey; DTP3, third dose of diphtheria-tetanus-pertussis vaccine.

access to skilled prenatal care has been linked to under-vaccination, or children not receiving all recommended vaccines,[15 43] and a prior study highlighted an association between the number of skilled healthcare workers (nurses and midwives) and DTP3 coverage using a cross-country analysis of developing countries around the world for the years 1990–2004.[40] Self-reported barriers limiting access to care for women when seeking treatment, namely travelling alone to healthcare facilities and travel distance, were associated with DTP3 vaccination. In regions with higher population density, women report fewer concerns regarding travelling alone to access care when sick. Rural communities report more challenges than urban communities with respect to accessing immunisation services. Unfortunately, retaining skilled health workers is more difficult in rural regions of Senegal.[44] Senegal

has addressed these challenges through training community health workers and health post staff, improving infrastructure for outreach services, increasing capacity for health post expansion, and improving surveillance and data management at local levels.[11] In 2011, a 5-year US$40 million grant from the United States Agency for International Development (USAID) helped Senegal construct, maintain and link more health huts to the national health system.[45]

Our results illustrate the role of contextual factors in relation to childhood vaccination. Although national childhood vaccination coverage in Senegal is relatively high compared with other LMICs, there are still gaps in coverage and disparities between regions which, similar to other studies, illustrate inequities in deployment of vaccination services.[12 13] National health programmes rely on

the consideration of contextual factors and heterogeneity between subnational areas, such as those identified in this study, to inform programmatic decision-making and improve health equity. Strategies for improving immunisation rates need to be tailored considering subnational differences; as stated in The Immunisation Agenda 2030, 'reaching all people will require higher national vaccination coverage, but also less subnational inequity'.[46] Subnational childhood vaccination has been negatively correlated with institutional mistrust.[47] Strong governance structures are required to prevent and address institutional mistrust while prioritising equity of healthcare access at the national level.[11 12 48] Subnational DTP3 coverage estimates have been compared with national health spending, GDP per capita and other governance indicators in sub-Saharan Africa to investigate the drivers of equitable immunisation services.[49] To reduce differences in vaccination coverage between regions, and address institutional mistrust, national health policies and programmes should consider heterogeneity between and within regions. Senegal has implemented top-down and bottom-up mechanisms for decision-making, planning and resource allocation to address subnational heterogeneity and allow for strategies and initiatives tailored for local populations.[11]

## Strengths and limitations

This work uses regionally representative data sets to explain associations between DTP3 coverage and demographic and health system characteristics at the subnational level within Senegal, and evaluates changes in vaccination coverage over time for each of these regions, considering the geographical location of each region. We focused on analysing respondent groups at the regional level or higher because DHS surveys were not sampled to be representative at the smaller district level. Heterogeneities at the subregional level were thus not evaluated. If district level data were available, we conjecture that negative associations between DTP3 coverage and other factors, such as poverty, could be identified, given heterogeneities at the subregional level. For instances where DTP3 immunisation was determined based on maternal recall, errors in recall could have biased results. Additionally, immunisation is only reported in the women's questionnaire of the DHS survey; bias may exist due to families of children with no mothers.

## CONCLUSIONS

Senegal has improved vaccination coverage across its 14 administrative regions despite high poverty and low female literacy as compared with other LICs and LMICs.[19] While coverage varies between regions, high coverage has expanded in recent years beyond coastal and urban regions into the inland and rural areas. National and regional efforts—especially increasing the number of health posts and huts to reach rural communities and expanding the community health worker programmes—have likely played an important role.[9] Historically, the inland and rural regions have had variability in coverage over time; future years will tell if the recent increases in coverage will be sustained.

**Acknowledgements** The authors thank the referees for thoughtful and constructive comments which significantly improved this paper. This research was supported in part by the following Georgia Tech benefactors: William W. George, Andrea Laliberte, Joseph C. Mello, Richard 'Rick' E. and Charlene Zalesky.

**Contributors** HKS, FC-Z, PK and DN contributed to study structure. HKS analysed DHS and national data and HKS, FC-Z, PK and DN synthesised key themes. ZMS, MCF and MS provided essential context relating to qualitative factors within Senegal. All authors contributed to drafting the article and approve of the final manuscript. HKS is the guarantor of this study and had full access to the data, controlled the decision to publish and accepts full responsibility for the conduct of this study.

**Funding** Funding for this study was provided by the Bill & Melinda Gates Foundation, OPP1195041.

**Map disclaimer** The inclusion of any map (including the depiction of any boundaries therein), or of any geographic or locational reference, does not imply the expression of any opinion whatsoever on the part of BMJ concerning the legal status of any country, territory, jurisdiction or area or of its authorities. Any such expression remains solely that of the relevant source and is not endorsed by BMJ. Maps are provided without any warranty of any kind, either express or implied.

**Competing interests** None declared.

**Patient and public involvement** Patients and/or the public were not involved in the design, or conduct, or reporting, or dissemination plans of this research.

**Patient consent for publication** Not applicable.

**Ethics approval** The current research was approved as exempt human research by the Institutional Review Board committee of Emory University, Atlanta, Georgia, USA (IRB00111474). All DHS Programme procedures and questionnaires are reviewed and approved by ICF Institutional Review Boards. The authors of this paper received permission from the DHS Programme for the use of the data. All DHS survey respondents remained anonymous to the researchers.

**Provenance and peer review** Not commissioned; externally peer reviewed.

**Data availability statement** Data are available on reasonable request. The data that support the findings of this study are available by requesting access at https://www.dhsprogram.com/data/available-datasets.cfm.

**ORCID iDs**
Pinar Keskinocak http://orcid.org/0000-0003-2686-546X
Moussa Sarr http://orcid.org/0000-0003-2372-6632
Matthew C Freeman http://orcid.org/0000-0002-1517-2572

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
