## [Reviewer comments · BMJ Open]

This paper was submitted to a another journal from BMJ but declined for publication following peer review. The authors addressed the reviewers' comments and submitted the revised paper to BMJ Open. The paper was subsequently accepted for publication at BMJ Open.

ARTICLE DETAILS

TITLE (PROVISIONAL)	Factors associated with vaccine coverage improvements in Senegal between 2005-2019: A quantitative retrospective analysis
AUTHORS	Smalley, Hannah; Castillo-Zunino, Francisco; Keskinocak, Pinar; Nazzal, Dima; Sakas, Zoë; Sarr, Moussa; Freeman, Matthew

VERSION 1 – REVIEW

REVIEWER	Bender Ignacio, Rachel University of Washington, Infectious Diseases/Medicine
REVIEW RETURNED	05-Jun-2023

GENERAL COMMENTS	This is a retrospective review of Demographic and Health Surveys as well as other national/regional level data to determine factors associated with vaccine coverage at a sub-national level in Senegal and changes over time. This is an important topic and this reviewer applauds the authors for highlighting the heterogeneity in vaccine uptake between regions and associated with various social determinants of health. The main criticism of this paper is that the DHS is not well described both in the data that it collects, but also the number of respondents was never mentioned in the main text (only in abstract and figure), but also does not seem to encompass all data collection reflecting the longitudinal analyses from 2005-2018 (the n=1183 seems to correspond to 2019?). Additionally, there are 2 sets of figures and 2 sets of references and their integration into the paper is not clear. Revision to the methods and results reporting would be important to help interpretation of this important data. Detailed feedback: -Background: Please define sub-nationally in general and for this purpose- are these geographic or political regions, provinces or other jurisdictions? Methods: -Need more detail please on the DHS. For example, in the abstract it states n=1183 living children. Was this performed for 1 child per household? or was the vaccination rate per household taken into account? (presumably it is only the index baby born in that year in that household, but not clear).
---

-Is that n per each year of the study from 2005-2019 or overall? Presumably with growing population and actual vs intended sampling the number might vary year to year with goal approximately 0.1% coverage (this only also stated in supplement)

-Part of the DHS assesses literacy- how was the sampling done to be representative and how was it administered, especially to families without literacy?

-Not specified, but it seems to indicate that DHS was only administered to mothers? what about families without mothers? were other guardians/family members assessed or were these households not included? It seems as though a valuable data point might be whether or not the mother was the caregiver/alive or if other family was the responsible party

-How long after birth was the survey administered and what was the window allowed for within window/late vaccination? Per WHO, dose #3 of DTP is given 14-18 weeks standard, but could potentially have 4 weeks grace period for catch up for example. The methods state Senegal expects #3 at 14 weeks, but methods should have further description of windows allowed for on-time and catch up vaccination.

-Was vaccine receipt only self report from the parent through DHS or official record? What method was done to verify, especially if DHS performed at 12-23 months, recall may not be complete (especially if low literacy)

-Please provide definition on how literacy was assessed and skilled prenatal care

-Supplement compares Senegal's coverage over time to other regional LIC/LMIC- The methods for comparison to other countries within the Exemplar report are only in the supplement- it is unusual to present both methods and results that appear only in the supplement rather than using supplement to augment methods or results presented in main manuscript. Extra-country comparisons not well discussed other than supplement.

-Please improve discussion in methods and results about how change over time was evaluated. Methods states independent ttests comparing average change over time (is this from 2005 to 2019? and by region? if comparing by time and region (Figure 4) needs ANOVA other test for multiple groups. Also, results discuss variance in coverage, which is somewhat different than change 2019- 2005 and would use a different statistical test. Please clarify.

-Presumably this data is all considered de-identified public health data, but a statement to this effect should appear at the end of the methods. The statement of "no patient or public involvement" does not necessarily respond to this usual statement.

Results:

-Please start the results section with univariate descriptive analyses, especially including how many households were surveyed in each year and characteristics overall for 2019 prior to discussing bivariate associations.

Please consider reviewing this article about challenging norms of publications involving collaborations between authors from LMIC and those from high resource countries/institutions in considering authorship and author order:

<https://gh.bmj.com/content/4/5/e001853>

Supplement: None of the methods or outputs (figures) are referenced in the methods or results as mentioned above. The figures use the same figuring as main figures in manuscript.

	Supplemental results should be referenced in the main manuscript and differentiated from main figures. The separate set of references for the supplement is also confusing.
--	---

REVIEWER	Seror, Valerie INSERM, U912
REVIEW RETURNED	26-Jun-2023

GENERAL COMMENTS	The manuscript focuses on childhood vaccination and inter-regional disparities in vaccine uptake. Using DTP3 as a proxy for estimating the retention of children in vaccine programs, this study based on using data from the DHS in 2005, 2010/2011, 2012/2013 and 2014/2019 had been conducted to identify the factors associated with DTP3 coverage in Senegal at both the household level and regionally. Statistical analysis had been conducted based on regional aggregated data (14 regions). Among the main findings, higher maternal literacy and access to skilled prenatal care were found to increase uptake, whereas traveling alone to the health post was identified as a barrier to getting vaccinated. A first main comment relates to the issue of identifying factors at both the household and regional level. The factors identified related to both provision of health services (availability and proximity) and households' decision-making to get DTP3 vaccine. As a result, these findings make it difficult to disentangle the impact of living in urban areas and associated easier access to education and healthcare from the impact at the household level. Another main comment relates to the data analysis. It is unclear to understand why the statistical analysis did not involve weighting the study samples using key variables in order to reach representativeness in the different regions. Table 1 could be moved to the Result section when filled to present the raw data used in the study.
--

VERSION 1 – AUTHOR RESPONSE

Reviewer: 1

Dr. Rachel Bender Ignacio, University of Washington, Fred Hutchinson Cancer Research Center

Comments to the Author:

This is a retrospective review of Demographic and Health Surveys as well as other national/regional level data to determine factors associated with vaccine coverage at a sub-national level in Senegal and changes over time.

This is an important topic and this reviewer applauds the authors for highlighting the heterogeneity in vaccine uptake between regions and associated with various social determinants of health. The main criticism of this paper is that the DHS is not well described both in the data that it collects, but also the number of respondents was never mentioned in the main text (only in abstract and figure), but also does not seem to encompass all data collection reflecting the longitudinal analyses from 2005-2018 (the n=1183 seems to correspond to 2019?). Additionally, there are 2 sets of figures and 2 sets of references and their integration into the paper is not clear. Revision to the methods and results reporting would be important to help interpretation of this important data.

Response: Thank you for your careful review of this manuscript. Responses to specific feedback are below. We have added a thorough description of DHS data, including the number of respondents across all data used for the longitudinal analyses from 2005-2018. We have moved the supplemental file references to the main manuscript, cited appropriately, and added clarification regarding the material presented in the supplement.

Detailed feedback:

-Background: Please define sub-nationally in general and for this purpose- are these geographic or political regions, provinces or other jurisdictions?

Response: The 14 regions discussed in the manuscript are administrative regions; each region is administered by its own governor. We have clarified what we mean by subnationally in the methods section of the text.

Methods:

-Need more detail please on the DHS. For example, in the abstract it states n=1183 living children. Was this performed for 1 child per household? or was the vaccination rate per household taken into account? (presumably it is only the index baby born in that year in that household, but not clear).

Response: We have added more detail on the DHS survey data in the methods section of the manuscript. The analysis considers all children in the 12-23 month age group for determining vaccination coverage. We have stated the following in the Methods section: "Child immunization is covered in the women's questionnaire in DHS surveys with responses representing each child of the women surveyed." We have also clarified how we filter the data (i.e., for living children ages 12-23 months), and we report the number of respondents meeting criteria for each survey year in Table 1 in the results section. Note that for each year of analysis (2005-2019), the percentage of responses corresponding to mothers with a 2nd child meeting the criteria is between 1.3%-2%.

-Is that n per each year of the study from 2005-2019 or overall? Presumably with growing population and actual vs intended sampling the number might vary year to year with goal approximately 0.1% coverage (this only also stated in supplement)

Response: We have moved Table 1 to the results section and edited it to report the number of survey respondents (unweighted) in each year for survey years from 2005-2019; breakdowns by region are reported in Table S1 in the supplemental materials. You are correct that the number varies year to year. The 0.1% coverage statement was erroneous – see response to next comment below.

-Part of the DHS assesses literacy- how was the sampling done to be representative and how was it administered, especially to families without literacy?

Response: We have provided the following details in the methods section of the paper with respect to how the survey samples are selected, and directed the reader to the necessary citations for learning more: "28 sampling strata are created which correspond to urban and rural parts for each of the 14 regions; samples are drawn independently from these sampling strata. Specific details on sampling methods for each survey year can be found in the appendices of each report (see [22])." A mother is considered to be literate if she is able to read a whole sentence or part of a sentence as reported in the DHS data and per DHS program recommendations (<https://dhsprogram.com/Data/Guide-to-DHS-Statistics/index.cfm>). There is a questioner who asks the respondent the survey questions. For example, for determining literacy, the questioner asks the respondent to read from a card. We have added this information into the methods section.

-Not specified, but it seems to indicate that DHS was only administered to mothers? what about families without mothers? were other guardians/family members assessed or were these households

not included? It seems as though a valuable data point might be whether or not the mother was the caregiver/alive or if other family was the responsible party

Response: DHS surveys households as well as women aged 15-49 and men aged 15-59. However, child health, and specifically immunization coverage, is only covered in the women's questionnaire in DHS surveys (https://dhsprogram.com/Methodology/Survey-Types/DHS-Questionnaires.cfm#CP_JUMP_16179). We have clarified this in the text. We agree that it would be valuable if this information were gathered from other family members and/or responsible parties as well but currently it is not.

-How long after birth was the survey administered and what was the window allowed for within window/late vaccination? Per WHO, dose #3 of DTP is given 14-18 weeks standard, but could potentially have 4 weeks grace period for catch up for example. The methods state Senegal expects #3 at 14 weeks, but methods should have further description of windows allowed for on-time and catch up vaccination.

Response: DHS are large-scale cross-sectional surveys. We consider mothers with living children aged 12-23 months in the analysis. By only considering cases with children aged 12 months or older, we are able to capture late vaccination cases. We do not address issues with on-time versus late vaccination (i.e., 14-18 weeks standard vs. within the grace period vs. late); rather we are concerned with whether or not the child has been vaccinated by the time of the survey if they are in this 12-23 months age group.

-Was vaccine receipt only self report from the parent through DHS or official record? What method was done to verify, especially if DHS performed at 12-23 months, recall may not be complete (especially if low literacy)

Response: We consider a vaccination as administered if it is reported by the mother or marked on the child's vaccination card with or without a date; this latter information is included in the DHS data. If not marked on the child's vaccination card and the mother did not recall, then the vaccination is assumed not to have been given.

-Please provide definition on how literacy was assessed and skilled prenatal care

Response: We have clarified the following in the text.

"Literacy is assessed in the survey by the questioner asking the respondent to read from a card. Prenatal (or antenatal) care is assessed by asking women who attended their last birth, with the following options: doctor, nurse/midwife, auxiliary nurse/midwife, community health worker, other health worker, traditional birth attendant, other, and no antenatal care."

"For our analysis, we consider a woman to be literate if she is able to read a whole sentence or part of a sentence as reported in the DHS survey. We consider a woman to have received skilled prenatal care if she answered "doctor", "nurse/midwife", or "auxiliary nurse/midwife" as attending her most recent birth."

-Supplement compares Senegal's coverage over time to other regional LIC/LMIC- The methods for comparison to other countries within the Exemplar report are only in the supplement- it is unusual to present both methods and results that appear only in the supplement rather than using supplement to augment methods or results presented in main manuscript. Extra-country comparisons not well discussed other than supplement.

Response: We have chosen to exclude this material from the supplement as these methods and figures are not crucial to the purpose of the paper. Instead we have referenced sources for the data supporting our arguments related to this material.

-Please improve discussion in methods and results about how change over time was evaluated. Methods states independent ttests comparing average change over time (is this from 2005 to 2019? and by region? if comparing by time and region (Figure 4) needs ANOVA other test for multiple groups. Also, results discuss variance in coverage, which is somewhat different than change 2019-2005 and would use a different statistical test. Please clarify.

Response: We have edited the text to clarify our methods. We clustered regions geographically into 2 groups (coastal and inland), calculated the variance in DTP3 values over time for each region, and performed independent sample t-tests to compare these two groups of variances.

-Presumably this data is all considered de-identified public health data, but a statement to this effect should appear at the end of the methods. The statement of "no patient or public involvement" does not necessarily respond to this usual statement.

Response: We have added a statement to this effect at the end of the methods section.

Results:

-Please start the results section with univariate descriptive analyses, especially including how many households were surveyed in each year and characteristics overall for 2019 prior to discussing bivariate associations.

Response: We have added a table to the beginning of the results section which reports the number of survey respondents meeting criteria in each year evaluated (Table 1) as well as more detailed information for year 2019 in the supplement (Table S2). We have also included in the supplement a table which reports survey respondents for each region (Table S1).

Please consider reviewing this article about challenging norms of publications involving collaborations between authors from LMIC and those from high resource countries/institutions in considering authorship and author order: <https://gh.bmj.com/content/4/5/e001853>

Response: Thank you for referring us to the article. We are aware of these sensitivities and have made efforts in this overall project (this is one of over a dozen papers) to openly discuss authorship, have an inclusive and transparent authorship process, and provide opportunities for substantive LMIC contributions. We have chosen our current authorship order based on various factors; aside from first and senior author, the authorship order is alphabetical.

Supplement: None of the methods or outputs (figures) are referenced in the methods or results as mentioned above. The figures use the same figuring as main figures in manuscript. Supplemental results should be referenced in the main manuscript and differentiated from main figures. The separate set of references for the supplement is also confusing.

Response: We have removed the figures previously presented in the supplement and added references to the tables in the supplement within the main body text. We have also relabeled the supplemental tables so they can be differentiated from the main tables (e.g., Table S1 etc.). We have also combined the references into a single set reported with the main manuscript.

Reviewer: 2

Dr. Valerie Seror, INSERM

Comments to the Author:

The manuscript focuses on childhood vaccination and inter-regional disparities in vaccine uptake. Using DTP3 as a proxy for estimating the retention of children in vaccine programs, this study based on using data from the DHS in 2005, 2010/2011, 2012/2013 and 2014/2019 had been conducted to identify the factors associated with DTP3 coverage in Senegal at both the household level and regionally. Statistical analysis had been conducted based on regional aggregated data (14 regions).

Among the main findings, higher maternal literacy and access to skilled prenatal care were found to increase uptake, whereas traveling alone to the health post was identified as a barrier to getting vaccinated.

A first main comment relates to the issue of identifying factors at both the household and regional level. The factors identified related to both provision of health services (availability and proximity) and households' decision-making to get DTP3 vaccine. As a result, these findings make it difficult to disentangle the impact of living in urban areas and associated easier access to education and healthcare from the impact at the household level.

Response: Thank you for your thoughtful review of this paper. Yes, it is intuitive that living in urban areas (with the associated easier access to education and healthcare) would likely result in higher DTP3 vaccine uptake. We have performed additional analyses of the individual-level factors, where we control for location of residence (urban). We find that the factors found to be significantly associated with DTP3 coverage in bivariate analysis are also significantly associated when controlling for residence location as well. We have edited the text to relay these new results with the following addition: "We find that significant associations hold for female literacy, skilled prenatal care, distance, and traveling alone when adjusting by type of place of residence (urban)." The results are also in the tables below; Table R1 presents the regression analysis results as they are presented in the paper, and Table R2 presents the regression analysis results when controlling for type of residence (urban).

Table R1: Regression analysis results for household-level factors associated with DTP3 vaccination.

	Estimate	Std. error	Statistic	P value	Confidence interval: lowerbound	Confidence interval: upperbound
Urban	0.611	0.253	2.42	0.0157	0.134	1.13
Concerns - Money	-0.339	0.210	-1.61	0.106	-0.756	0.0685
Concerns - Permission	-0.0229	0.322	-0.0712	0.943	-0.616	0.655
Concerns – Distance	-0.697	0.209	-3.33	0.000866	-1.11	-0.283
Concerns - Alone	-1.08	0.218	-4.94	0.000000785	-1.50	-0.644
Skilled Prenatal Care	1.80	0.359	5.00	0.000000568	1.06	2.48
Female Literacy	1.11	0.326	3.39	0.000704	0.509	1.80

Table R2: Regression analysis results for household-level factors associated with DTP3 vaccination when controlling for type of residence (urban).

	Estimate	Std. error	Statistic	P value	Confidence interval: lowerbound	Confidence interval: upperbound
Concerns – Money	-0.264	0.212	-1.24	0.214	-0.686	0.148
Concerns - Permission	0.0324	0.323	0.100	0.920	-0.564	0.713
Concerns - Distance	-0.587	0.218	-2.69	0.00711	-1.01	-0.157
Concerns - Alone	-0.991	0.224	-4.42	0.00000983	-1.43	-0.546
Skilled Prenatal Care	1.73	0.361	4.79	0.00000167	0.995	2.42
Female Literacy	1.01	0.330	3.06	0.00219	0.407	1.71

Another main comment relates to the data analysis. It is unclear to understand why the statistical analysis did not involve weighting the study samples using key variables in order to reach representativeness in the different regions.

Response: The DHS data is analyzed using sampling weights to reach representativeness in the different regions, but you are correct that this was not explicitly stated in the manuscript. The individual weight for women was utilized in the calculations of individual-level factors. Details are available at <https://dhsprogram.com/Data/Guide-to-DHS-Statistics/index.cfm>. We have added the following text:

“Data must be weighted because the overall probability of a household being selected is not constant across regions and residence types [28]. Thus, we compute the individual weight for women and households to adjust for differences in probability of selection and response between cases in a sample.”

Table 1 could be moved to the Result section when filled to present the raw data used in the study.

Response: We have chosen to edit and move Table 1 to the beginning of the results section; the table now specifies the number of survey respondents which meet criteria in each DHS survey year; more details for survey year 2019 are presented in the supplement (Table S2).

VERSION 2 – REVIEW

REVIEWER	Seror, Valerie INSERM, U912
REVIEW RETURNED	31-Aug-2023
GENERAL COMMENTS	Thank you for your detailed answers, I have no other comment to address.